# Tooth Loss and Caries Experience of Elderly Chileans in the Context of the COVID-19 Pandemic in Five Regions of Chile

**DOI:** 10.3390/ijerph20043001

**Published:** 2023-02-09

**Authors:** Víctor Beltrán, Marco Flores, Cristina Sanzana, Fernanda Muñoz-Sepúlveda, Eloy Alvarado, Bernardo Venegas, Juan Carlos Molina, Sandra Rueda-Velásquez, Alfredo von Marttens

**Affiliations:** 1Clinical Investigation and Dental Innovation Center (CIDIC), Dental School and Center for Translational Medicine (CEMT-BIOREN), Universidad de La Frontera, Temuco 4811230, Chile; 2Postgraduate Program in Oral Rehabilitation, Interuniversity Center for Healthy Aging, Universidad de La Frontera, Temuco 4811230, Chile; 3Institute for Research in Dental Sciences, Faculty of Dentistry, Universidad de Chile, Santiago 8380544, Chile; 4Program of Master in Dental Sciences, Dental School, Universidad de La Frontera, Temuco 4811230, Chile; 5Department of Industrial Engineering, Universidad Técnica Federico Santa María, Santiago 7630000, Chile; 6Carlos Van Buren Hospital of Valparaíso, Valparaíso 2340000, Chile; 7Clínica MEDS, Santiago 8150513, Chile; 8Department of Oral Pathology, Faculty of Dentistry, Universidad Santo Tomás, Bucaramanga 680001, Colombia; 9Department of Prosthesis, Faculty of Dentistry, Interuniversity Center for Healthy Aging, University of Chile, Santiago 8380000, Chile

**Keywords:** tooth loss, caries experience, edentulism, elderly

## Abstract

Risk factors associated with tooth loss have been studied; however, the current status of the epidemiological profiles and the impact of the pandemic on the oral health of the elderly is still unknown. This study aims to determine the experience of caries and tooth loss among elderly Chilean citizens in five regions and to identify the risk factors associated with tooth loss. The sample includes 135 participants over 60 years old assessed during COVID-19 lockdown. Sociodemographic variables such as education and RSH (Social Registry of Households) were obtained through a teledentistry platform called TEGO. The history of chronic diseases such as diabetes, obesity, depression and dental caries reported by DMFT index scores were incorporated. The statistical analysis included Adjusted Odds Ratios (ORs) to assess risk factors associated with the lack of functional dentition. Multivariate hypothesis testing was used to compare the mean equality of DMFT and its components between regions (*p*-value < 0.05). Individuals with RSH ≤ 40% were at higher risk of having no functional dentition with OR 4.56 (95% CI: 1.71, 12.17). The only mean difference between regions was the filled tooth component. Tooth loss was associated with multidimensional lower income, where the elderly belonging to the 40% most vulnerable population had a higher prevalence of non-functional dentition. This study highlights the importance of implementing a National Oral Health Policy that focuses on oral health promotion and minimally invasive dentistry for the most vulnerable population.

## 1. Introduction

The world population is experiencing an ageing process. The World Health Organization (WHO) declared that between 2015 and 2050, the percentage of the planet’s inhabitants over 60 years of age will almost double from 12% to 22% [1]. Chile will no longer be the exception. According to the projections of the Chilean Ministry of Health (MINSAL), by 2025, Chile will be the oldest country in the region [2]. This increase in the population’s life expectancy constitutes a great challenge for the health profession, given the high burden of oral disease and its relation to general health.

Tooth loss reflects the endpoint of a lifetime of dental disease and the individual’s history or absence of dental treatment [3]. To prevent the consequences of edentulism the WHO, the World Dental Federation (FDI), and the International Association for Dental Research (IADR) urged countries on the importance of reducing the number of elderly edentulous people in ages ranging from 65 to 74 years. This effort has permitted an increase in the number of natural teeth present in the oral cavity and the number of individuals with functional dentitions [4], which has meant a significant decline in the prevalence and incidence of total tooth loss worldwide [5]. In Chile, more than half of the elderly population has 11 or less teeth, and the number of senior citizens over 65 years with no functional dentition remains elevated; in fact, 81.7% have less than 20 natural teeth [6]. One way of reflecting the success of the public policies taken to diminish the number of elderly people experiencing tooth loss and caries is assessing the DMFT index [7], which summarizes the number of decayed teeth (DT), missing teeth(MT) and filled teeth (FT) to reflect the actual disease experience (past and present), and, if evaluated separately, indicates the management of the disease [8].

Maintaining optimal oral health by preserving a natural, healthy and functional dentition contributes to the survival of older adults [9], since missing unreplaced teeth has been associated with an increase in the risk of malnutrition, frailty and cardiovascular mortality [10]. Likewise, an association with the decline in cognitive function, dementia, obesity, diabetes and hypertension has been reported [11,12,13,14,15]. In this sense, the WHO’s Oral Health Program has encouraged national planners to strengthen the implementation of systematic programs aimed at improving healthy aging, oral health and a better quality of life for the elderly [16]. However, to this day, the neglect of oral health constitutes a failure of global health policies in delivering the basic human rights of older people [17]. In the context of the pandemic, due to the restrictions to prevent contagion by COVID-19 and the serious consequences this disease can bring to the elderly population in particular, measures such as social distancing and limitations on the number of face-to-face visits to access routine medical and dental check-ups were taken [18,19,20], so it is expected that the deficient general and oral health in these people would have become more severe than before the pandemic [21].

To provide a response to dental emergencies and priority treatments in the context of the COVID-19 pandemic, a technology based on teledentistry concepts called Tele-platform of Geriatric and Dental Specialties (TEGO) was developed. The pilot test of this web platform showed that a semi-presential teledentistry workflow can help elderly people who are impeded to look for traditional dental assistance during a pandemic, and made it possible to obtain a database with relevant information from the medical-dental examination carried out on the elderly population in five regions of Chile [22]. Former studies and National surveys related age, educational level, low income, depression and oral health inequalities as the main risk factors of dental caries and tooth loss in Chilean adults [23,24,25]. These studies are crucial due to the continuous need to provide adequate information for decision making. The objective of this study is to determine the experience of caries and tooth loss among the Chilean population over 60 years of age in five regions of Chile and identify the risk factors associated with tooth loss, aiming to update local epidemiological information. Our hypothesis assumed that despite the current pandemic context, we could expect a better state of oral health in the population studied.

## 2. Materials and Methods

### 2.1. Sampling Methodology

In Chile, the elderly segment of the population is defined as people of 60 years and over [26]. The target population of this study were males and females aged 60 years or older, living in the regions of Antofagasta, Metropolitana, Maule, Bio Bio and La Araucanía. The national service for the elderly (SENAMA) promotes healthy ageing and equal rights for the elderly by articulating intersectoral networks in which groups and clubs of the elderly participate [27]. We recruited participants through databases of SENAMA regional coordination and community organizations between 2020 and 2021. We chose these regions due to the feasibility of implementing the teledentistry platform in these geographical areas, since most of them were near the universities that contributed to the development of this research. Within each region, we followed a stratified sampling design to select the specific districts that were to be included in the study. In this context, the allocation for each region was set proportionally to the resident population according to the projections made by the National Statistics Institute of Chile (INE). A final count of 135 participants were included in this study. This study was approved by the Scientific Ethics Committee of the Universidad de La Frontera, Temuco, Chile (Folio Number 090/20).

### 2.2. Selection of Study Participants

A new teledentistry workflow protocol proposed by the research team of this article [28] was executed, for which the recruitment and admission of subjects was carried out by a social worker, who synchronously collected basic socio-demographic data via phone calls, performed a COVID-19 triage, and selected the participants for this study using the inclusion criteria. The inclusion and exclusion criteria in the present study were as follows:

Inclusion criteria:Older adult (over 60 years);Dental urgency or requirement of priority care:
°Severe dental pain that does not yield to analgesics.°Recent trauma. Direct blow that involves teeth or mouth, accompanied by severe pain.°Oral bleeding.°Significant swelling of the mouth, face or neck.°Stains or wounds in any part of the mouth that do not disappear in a month.°Loss or fracture of restorations or dental prostheses.°Injuries to the mucosa, due to dental prosthesis mismatch.°Dental treatment required prior to urgent critical medical procedures.

Exclusion criteria:Anticoagulant therapy;Chronic diseases without treatment;Cancer treatment;Dialysis.

### 2.3. Data Collection

All subjects agreed to participate in the present study by signing an informed consent. Information was recorded on the platform named Geriatric Dental Specialties Teleplatform (TEGO by its acronym in Spanish: “Teleplataforma de Especialidades Geriátrico Odontológicas”) [29]. This web-based platform integrates anamnesis modules, a novel 3D standardized model for indexing relevant information for each case and also allows teleconsultations with specialists.

Once the participants had been selected, full sociodemographic data were collected, including the Social Registry of Households in terms of percentile ranges (RSH) [30] and educational level (classified as complete basic education or less and incomplete secondary education or higher). The Barthel index [31] and an OHIP-14SP quality of life questionnaire [32] were applied.

After the participants had been scheduled on the TEGO platform, the face-to-face care of the recruited subject was carried out by a general dentist in a mobile dental unit. The face-to-face care was carried out by three different general dentists with experience in primary health care, who carried out a complete medical-dental-geriatric anamnesis, which included the following history of chronic diseases: diabetes (obtained from the participants’’ reports), obesity (the subjects were measured and weighed inside the mobile clinic), and depression (the shortened Yesavage geriatric depression scale was applied [33]).

Then, the dentist performed a complete general physical and dental examination and the information was recorded in a complete electronic clinical record on the TEGO platform. The dental examination included the application of the DMFT index scores that were recorded following WHO recommendations, where the M component comprises missing teeth due to caries or for any other reason [34]. The data were collected between February and June of 2021.

## 3. Results

Using age categorization [34], frequently used in studies of this nature, Table 1 shows the absolute and relative frequencies of the 135 participants included in the research, differentiated by sex. A greater participation of women was shown in the study (64.4%).

Table 2 shows the DMFT index together with its 95% confidence interval in parentheses. Additionally, t-student hypothesis tests were performed for the mean DMFT between both genders. No significant differences were found (*p*-value 0.6687).

In Table 3, we present the Odds Ratios (ORs) with their respective confidence intervals and *p*-values of the univariate and multivariate logistic regression, including educational level (8 years of studies or more), RSH* according to variables mainly related to the economic income of the household or those variables that seek to reflect the income of a household [30]. For this study, subjects were classified as follows: RSH that corresponded to the up to 40% more vulnerable population and the population over this segment (greater than 40%), and the presence of diabetes, depression, or obesity. Following a similar methodology to Urzúa et al. [23], our analysis established RSH* as the only statistically significant variable to predict the presence of tooth loss.

Of the elderly participants, 8.1% were edentulous. Table 4 shows the mean of the DMFT index by region and its respective breakdown into Decayed (D), Missing (M) and Filled (F) Teeth. The average for each region of Chile is shown in the age column.

In order to determine if there is a significant statistical difference in the mean DMFT and their component between the different regions of Chile, multivariate hypothesis testing was performed, namely Wilks Lambda, Pillai’s Trace, Lawley–Hotelling Trace and Roy’s Largest Root showing there is not enough statistical evidence to infer that the mean DMFT and their components of Decayed (D) and Missing (M) teeth are different between regions. Nonetheless, the opposite conclusion was obtained for the Filled Teeth (F) component of the DMFT index (*p*-value 0.0268).

## 4. Discussion

DMFT studies provide relevant information about emerging preventive dental practices in each country’s society [7,35,36,37,38,39]. The findings concerning the oral health of the elderly population (60 years and older) treated as part of our study in five regions of the country were similar to that observed in the Chilean adult population. The DMFT index (Table 2 and Table 4) was likewise similar to studies carried out in our country [23,40,41], demonstrating that the high caries experience of this group remains. Older adults with an age range of 65–74 years had DMFT scores (21.9) similar to those reported by Urzúa et al. (21.57) [23] and Mariño et al. (21.6) [40], and lower than those observed by Quinteros et al. (25.68) [39,41].

If we compare these data with other international studies under the same age segmentation, particularly from Spain [42], we observe that the rate of caries experienced in this European country is lower than that obtained in our study (16.38). In contrast with studies carried out in other Latin-American countries such as Brazil [43] and Uruguay [44], we found a higher rate than in Chile (29.24 and 24.1, respectively). Similarly, when compared with a study carried out in Mexico [45] under the same age segmentation, we found that the index of our study is similar to that obtained in that country (20.1).

At a global level, an increase in dental retention has been reported [5] and according to the WHO, public health policies in the elderly population should lead to the improvement of the tooth loss index, which, in our sample, was 14.74, measured by mean MT (Table 2), being similar to the study by Urzúa et al. (17.46) [23] and Mariño et al. (17.9) [40]; much lower than Quinteros et al. (22.36) [41]. Since tooth loss has been associated with sociodemographic characteristics, as well as with individual factors such as lifestyle and general health [46,47,48,49,50], the difference between these studies could be explained due to the different sample sizes used in each study, in addition to the differences in the target population, among other factors. In relation to the rate of edentulism, a lower prevalence is observed (8.1%) when compared with previous national studies that varied between 11.4% and 25% [6,23,40,51]. The prevalence detected may be lower than it is currently due to the objective of this project, which is not sought by edentulous patients.

In relation to tooth loss, Urzúa et al. [23] studied the factors of educational level, personal and family income, and presence of depression and obesity and related it to functional tooth loss, finding family income as the only statistically significant variable within the context of logistic regression analysis. The same factors were incorporated in our study, except for the variables related to economic income, in which we used the Socioeconomic Classification (CSE) of the Household Social Registry (RSH), which attempts to measure socioeconomic vulnerability in different dimensions. This official classification is used by the Chilean government to support the selection processes of beneficiaries within a wide range of subsidies and social programs [30]. The Socioeconomic Classification (CSE) places each household in in one of seven segments related to their income: 0–40%; 41–50%; 51–60%; 61–70%; 71–80%, 81–90% and 91–100%. Each section reflects the level of vulnerability in the population, with the first section being the most affected. A model based only on family monetary income would place households in the same section of the CSE, not considering the number and the characteristics of the members within the household; for this reason, the RSH classification is more complete to determine the degree of vulnerability. Therefore, the variable included in the regression analysis refers to people that belong in Section 1 of this categorization. In our study, belonging to the most vulnerable group (up to 40%) was found to be the only statistically significant variable to describe the presence of tooth loss (*p*-value 0.002).

To evaluate if there is a different burden of caries experience among the regions of Chile, hypothesis tests of mean equality of the DMFT and its components were performed. Our study found that there is not enough statistical evidence to assert that the means of DMFT, Decayed (DT) and Missing (MT), are different. However, this was not the case for the Filling component (FT). This difference could be explained by the great inequality that exists in the restorative treatment of dental caries (FT) in comparison with caries experience (DMFT) [8]. In Chile, as well as in other countries [52], the health status of people with a lower income is reported as worse and with more physical limitations [53]. Furthermore, they are faced with inequities in the access to dental health provided by the public sector [54], being even more serious regarding older adults, where 9 out of 10 are part of the public national health insurance system (FONASA) [55]. The relationship between general and oral health has already been established [56,57] through the convergence of risk factors such as sociodemographic, access to health, lifestyle, and others. The prevention and control of non-communicable diseases (NCDs), such as oral diseases, are integral parts of the COVID-19 response due to its association with the severity and fatality rates of COVID-19 [58]. Regarding this matter, one study compared the number of admissions for urgent dental care between the different periods of the pandemic, reporting a decrease in the use of services during the lockdown and the second wave [59]. To our knowledge, in our country, there is a lack of studies that evaluate the differences in the use of dental services prior to and during the pandemic; however, a survey was conducted before our study, reporting that 44% of adults have had a dental problem during the pandemic; of these, 62% indicated that their problems have not been resolved due to fear of contagion (33%) and lack of clinical hours (18%) [60]. This evidence and the fact that this study was carried out during the second wave, which occurred between the months of April and June 2021 [61], supports the idea that this modality of teledentistry was an important tool to implement during the pandemic. To this end, this project sought to provide care to a group with limited dental access during the COVID-19 lockdown and performed a comprehensive analysis of older people through the evaluation of sociodemographic factors and a complete medical-dental-geriatric anamnesis. Thus, it should be noted that although our DMFT scores are in line with what has been observed in several other countries, they are still very similar to those reported almost a decade ago in Chile [23,40,41], remaining very high in a global context and indicating a high burden of disease throughout the individual’s lifetime, since the main component of the DMFT corresponded to the MT [38].As expressed by Quinteros et al. [41], this could be explained by not being beneficiaries of the oral health promotion and prevention activities that are currently available, such as the project in which this study is contextualized.

Even though our study did not find other associations with a lack of functional dentition through the logistical regression of our data, systematic and longitudinal studies related tooth loss with obesity [13,62,63], depression [64] and diabetes [14]. It is noteworthy that the prevalence of socioeconomic vulnerability, whether measured in net terms (personal or family economic income) or multidimensional (RSH), continues to be a constant in studies that measure the severity of tooth loss. It has also been said that in most countries, the health policies adopted seem inadequate to reduce the inequalities in oral health between socially vulnerable people [65]. Along with this, considering that Chile showed the lowest number of remaining teeth and the highest association with social inequity when comparing the prevalence of tooth loss with developed countries [66], it becomes necessary to adapt the former policies to new ones that focus on promotion, prevention and minimally invasive treatments. These activities must be accompanied by a strong focus on the most vulnerable strata in this age group, and should also incorporate the use of new technologies and mobile dental services in order to facilitate their access [22,67,68,69,70,71,72]. It is worth promoting further studies if the current health status of this age group is maintained due to the limited targeting of preventive public policies or in case it has worsened due to the current context of the pandemic; it has been hypothesized that loneliness and a lack of dental care could worsen the oral health of vulnerable groups [21].

Notwithstanding the relevance of finding the link described above, our study had some limitations that must be considered. Due to the confinement context, an exploratory cross-sectional study was applied, so caution must be exercised in the interpretation of the results, considering the relationship of these factors as a statistical association and not a causality. In addition, the subjects included come from regions selected by their convenience and cannot be extrapolated to the entire nation. These limitations emphasize the need to conduct a longitudinal representative survey including a larger number of regions from Chile. To our knowledge, this is the first study that relates the Social Registry of Households in terms of percentile ranges with oral health and that also compares the management of elderly individuals’ dental caries across regions. Another strength of this study is the evaluation of the oral health status of a highly vulnerable group through DMFT in a context in which a greater progression of oral diseases was projected.

## 5. Conclusions

As in other studies, tooth loss was associated with sociodemographic characteristics such as multidimensional lower income expressed in RSH, in which the elderly belonging to the 40% most vulnerable had a higher prevalence of non-functional dentition.Updating the epidemiological databases is crucial to provide evidence that supports the implementation of programs and public policies focused on the needs of the population. If we want to diminish the burden of tooth loss, the implementation of a National Oral Health Policy should be provided, centering on health promotion and minimally invasive dentistry, considering the most vulnerable groups as an integral part of it.Through the use of teledentistry during the SARS-CoV-2 pandemic, we were able to obtain data from this vulnerable population, which has limited access to public dental health. Therefore, we suggest for this approach to be applied both in research and in day-to-day consultations.We recommend that a similar longitudinal study be replicated on a larger scale, including a larger number of regions from Chile and other countries.

## Figures and Tables

**Table 1 ijerph-20-03001-t001:** Frequency of participants by age groups in years (relative percentage in parentheses).

Sex	60–64 Years	65–74 Years	75 Years and More	All
Female	17 (85)	42 (67.75)	28 (52.83)	87 (64.4)
Male	3 (15)	20 (32.25)	25 (47.16)	48 (35.6)
Total	20	62	53	135

**Table 2 ijerph-20-03001-t002:** Mean DMFT index and its components (95% confidence interval in parentheses).

Age Groups (Years)	DMFT	D	M	F
60–64	21.10 (18.45–23.74)	1.65 (0.95–2.34)	14.25 (10.61–17.88)	5.20 (3.55–6.84)
65–74	22.26 (20.97–23.54)	1.48 (1.07–1.89)	14.74 (12.84–16.63)	5.67 (4.51–6.84)
75 and more	22.40 (21.20–23.59)	1.92 (1.32–2.52)	15.73 (13.72–17.73)	4.75 (3.54–5.95)

**Table 3 ijerph-20-03001-t003:** Non-Adjusted and Adjusted odds ratios (ORs) and 95% confidence interval (95% CI) with the respective *p* values related to tooth loss (the presence of less than 21 teeth in mouth) for the ≥60-year-old group.

	Variables	Univariate OR (95% CI)	*p* Value	Multivariate OR (95% CI)	*p* Value
Educational	Complete basic or less	1.42 (0.54–3.72)	0.472	1.37 (0.5–3.73)	0.542
	Incomplete medium or higher	1.0 (ref)			
RSH *	Up to the 40% more vulnerable	4.55 (1.79–11.59)	**0.001**	4.56 (1.71–12.17)	**0.002**
	Greater than 40%	1.0 (ref)			
Diabetes	YES	0.86 (0.32–2.28)	0.760	0.76 (0.27–2.13)	0.602
	NO	1.0 (ref)			
Depression	YES	0.88 (0.36–2.26)	0.795	0.79 (0.28–2.24)	0.658
	NO	1.0 (ref)			
Obesity	YES	0.62 (0.25–1.50)	0.287	0.86 (0.32–2.34)	0.768
	NO	1.0 (ref)			

* RSH—Social Registry of Households in terms of percentiles ranges.

**Table 4 ijerph-20-03001-t004:** DMFT index and its components (mean and SD) by Chilean regions.

Region	N	Age	DMFT	D	M	F
Antofagasta	12	71.33 (5.58)	20 (5.83)	1.41 (0.90)	11.58 (6.63)	5.33 (4.03)
Metropolitana	79	73.11 (6.30)	22.41 (5.01)	1.64 (2.08)	14.34 (6.63)	6.02 (4.58)
Maule	11	72 (6.18)	21.81 (4.09)	2.27 (2.14)	16.54 (7.01)	3 (3.94)
Bio Bio	18	71.11 (5.37)	22.61 (4.48)	1.61 (1.24)	16.05 (8.34)	4.94 (4.43)
La Araucanía	15	70.33 (8.26)	22.6 (5.61)	1.6 (1.63)	18.026 (7.50)	2.73 (2.71)
Total	135	72.28 (6.35)	22.21 (4.99)	1.66 (1.85)	14.94 (7.60)	5.20 (4.41)

## Data Availability

Not applicable.

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
