# Peer review of "Tooth Loss and Caries Experience of Elderly Chileans in the Context of the COVID-19 Pandemic in Five Regions of Chile"

_ijerph, 2023, doi:10.3390/ijerph20043001_

Round 1

Reviewer 1 Report

This is a small but complex study of certain components of the oral health drawn from selections of older persons from Chile. The findings confirm previous publication of findings of associations between income and oral health inequities, both in Chile and internationally. The authors should clarify the points below:

1. The title of the paper does not clearly represent the aims of the study.  For example, the study sample was drawn from 5 regions of Chile but was not a randomized sample but one of convenience (see page 2 lines 92-94). Further, the regions were chosen because of the "feasibility of implementing" the teledentistry platform. The title implies that the findings are generalizable to the national "Chilean Elderly" and does not mention the focus of the teledentistry program. Further, the title refers to the context of the COVID-19 pandemic yet there is no presentation nor analysis of specific data related to the subjects' use of dental services during the pandemic vis-a-vis prior services usage.

2. The literature reviewed identifies the high rate of edentulism in Chilean older persons, yet the data provided in the results do not identify the proportion of subjects who had no natural teeth (edentulous rate).

3. The basic methods and statistical analyses appear appropriate - although the sample size is clearly small. Further, it is not clear which variables (and how many) entered into the Adjusted OR analyses as compared with the Univariate OR.  Nor is there any comment regarding the calibration of the three dental examiners who performed the mouth examinations.  

4. The results are presented in five Tables with Table 5 being essentially redundant and could simply be presented within the text.  The only finding being that the F-component of the DMFT varied across regions. As mentioned above - total tooth loss was not considered as a unit outcome measure.

5. The Discussion is extensive given the narrow range of findings and does not describe the strengths, and more importantly, the weaknesses of the study.  A paragraph would be essential regarding this issue.

6. Advocacy for the use of teledentistry, especially during the conditions of a pandemic, was featured in the conclusion; however, the hypothesis that teledentistry would/could improve service provision/access was not specifically tested as a clear objective of the study.  It was simply a tool used in the survey. Clarification around this issue is necessary.

7. There are minor grammatical and typographical errata that need correcting eg "The world population is experiencing an aging process", the DMFT T-component should be referred to as Teeth not Tooth, spelling of teledentistry (line 93 page 2), patients should be replaced by the term subjects or participants, and the format of the Tables should be reviewed in the light of the Journal requirements

Author Response

Dear reviewer,

Thank you very much for your kind comments and corrections, in relation to these we have worked on the following modifications which we hope to be well received:

  1. Point 1: The title of the paper does not clearly represent the aims of the study.  For example, the study sample was drawn from 5 regions of Chile but was not a randomized sample but one of convenience (see page 2 lines 92-94). Further, the regions were chosen because of the "feasibility of implementing" the teledentistry platform. The title implies that the findings are generalizable to the national "Chilean Elderly" and does not mention the focus of the teledentistry program. Further, the title refers to the context of the COVID-19 pandemic yet there is no presentation nor analysis of specific data related to the subjects' use of dental services during the pandemic vis-a-vis prior services usage.

 Response 1: Regarding your comment stating that “The title of the paper does not clearly represent the aims of the study” we have come to the conclusion that we agree with your suggestion since it is not possible to generalize Chile as a country because the regions were not randomly selected. However, among the selected regions due to their associated ease in implementing the platform, the sampling methodology was randomized, which makes the results representative, but only of the selected regions. For that reason, we have modified the title and objective of the article.

Regarding mentioning the focus on the teledentistry program, we would like to state that that was not our aim, but a tool used to carry out this study. Nevertheless, we have included lines 149 to 153 describing the characteristics of the TEGO tool and its usefulness is mentioned in lines 86 to 90.

About the context of the COVID-19 pandemic, we have included some comments in our discussion (lines 290 to 300) in which we mention a comparison in the use of dental services before and during the pandemic.

  1. Point 2: The literature reviewed identifies the high rate of edentulism in Chilean older persons, yet the data provided in the results do not identify the proportion of subjects who had no natural teeth (edentulous rate).

Response 2: Thanks a lot for your comment. The high rate of tooth loss in the Chilean elderly population from the five regions included in the study appears on table 4, as part of the DMFT index, precisely on the component M. We have included the rate of edentulism in line 207 and we discussed this results in lines 251 to 254.

  1. Point 3: The basic methods and statistical analyses appear appropriate - although the sample size is clearly small. Further, it is not clear which variables (and how many) entered into the Adjusted OR analyses as compared with the Univariate OR.  Nor is there any comment regarding the calibration of the three dental examiners who performed the mouth examinations.  

Response 3: Thanks a lot for your comment. None of the variables incorporated in our logistical regression model are examiner-dependent. The incorporated variables were: educational level, RSH, diabetes, depression and obesity. However, taking this comment in consideration, we have modified Table 3. These changes can be seen on line 191-192, 202, and in the table itself.

  1. Point 4: The results are presented in five Tables with Table 5 being essentially redundant and could simply be presented within the text.  The only finding being that the F-component of the DMFT varied across regions. As mentioned above - total tooth loss was not considered as a unit outcome measure.

Response 4: Thanks a lot for the suggestion. We have thus eliminated Table 5 from the manuscript.

  1. Point 5: The Discussion is extensive given the narrow range of findings and does not describe the strengths, and more importantly, the weaknesses of the study.  A paragraph would be essential regarding this issue.

Response 5: We appreciate your suggestions and following on these we have added a paragraph between lines 333 and 344 in which me mention the strengths and weaknesses of the study.

  1. Point 6: Advocacy for the use of teledentistry, especially during the conditions of a pandemic, was featured in the conclusion; however, the hypothesis that teledentistry would/could improve service provision/access was not specifically tested as a clear objective of the study.  It was simply a tool used in the survey. Clarification around this issue is necessary.

Response 6: Thanks a lot for your comment. In relation to this, we’d like to mention that the main objective of our study was to determine the experience of caries and tooth loss among the Chilean population over 60 years of age of five region of Chile and identifying the risk factors associated with tooth loss, aiming at updating local epidemiological information. The Teledentistry Platform was used to achieve this goal working as a tool. However, to clarify this point, we have provided more details regarding the platform, and we have also indicated the platform’s functionalities (lines 86 to 91 and 149 to 153).  

  1. Point 7: There are minor grammatical and typographical errata that need correcting eg "The world population is experiencing an aging process", the DMFT T-compone.nt should be referred to as Teeth not Tooth, spelling of teledentistry (line 93 page 2), patients should be replaced by the term subjects or participants, and the format of the Tables should be reviewed in the light of the Journal requierement

Response 7: We appreciate how thorough you review has been. Based on your suggestions, we have performed the following changes: On line 66 and 67 of the manuscript we have changed the word “tooth” for “teeth”. Regarding our spelling of teledentistry, we have noticed that this spelling has been widely used and has found its place in dental terminology. The term “patient(s)” has been replaced following your suggestion. The tables have been reviewed.

Reviewer 2 Report

The study is genuine and very interesting. The manuscript is well written, however the authors are required to do the following changes to improve the quality of the manuscript:

- The authors should add a short statement in the abstract to show the current research gap and question.

- Research hypothesis/hypotheses should be added to the last section of the manuscript.

- The period of data collection should be added in the material and method section.

- The discussion section should include the limitation(s) of this study and directions for future research projects.

- The conclusions may be summarized and written in bullet points.

Author Response

Dear reviewer,

Thank you very much for your kind comments and corrections, in relation to these we have worked on the following modifications which we hope to be well received:

Point 1: The authors should add a short statement in the abstract to show the current research gap and question.

Response 1: We thank you for your suggestions and based on that we have included a brief statement in the summary (lines 27 to 28) showing the gap and the current research question.

Point 2: Research hypothesis/hypotheses should be added to the last section of the manuscript.

Response 2: We thank you for this correction and taking it in consideration, we have added the hypothesis in the last section of introduction on line 98-99 and we have discussed rejecting it throughout the discussion

Point 3: The period of data collection should be added in the material and method section.

Response 3: We thank you for this correction and taking it into consideration, we have added the data collection period on line 171 of the Materials and Methods Section.

Point 4: The discussion section should include the limitation(s) of this study and directions for future research projects.

Response 4: In relation to your suggestion, we have included a paragraph on the discussion (lines 328 to 345) in which the limitations of the study are mention, together with recommended further studies on this line of research.

Point 5: The conclusions may be summarized and written in bullet points.

Response 5: Thanks a lot for this suggestion. The conclusions have thus been summarized and written in bullet points as can be corroborated from lines 347 to 362.

Reviewer 3 Report

Can you please also specify the initial pool of patients from which the 135 patients were selected?

What are [AF1] - line 158, [AF2] - line 167, [fm3] - line 167, [AF4] - line 191, [AF5] - line 204, [AF6] - line 209, [AF7] - line 250?

References [49], [50] and [51] from lines 235-237 should not be exponential.

The conclusion section could be expanded more.

Author Response

Dear reviewer,

Thank you very much for your kind comments and corrections, in relation to these we have worked on the following modifications which we hope to be well received:

Point 1: Can you please also specify the initial pool of patients from which the 135 patients were selected?

Response 1: Regarding the initial group of subjects, these were elderly patients who presented dental urgencies or required priority dental attention during the years 2020 and 2021. These patients also belonged to neighborhood associations for the Elderly (SENAMA). A paragraph has been added to Materials and Methods that describes more precisely this initial group of patients (lines 106 to 114).

Point 2: What are [AF1] - line 158, [AF2] - line 167, [fm3] - line 167, [AF4] - line 191, [AF5] - line 204, [AF6] - line 209, [AF7] - line 250?

Response 2: We are sorry we could not understand this suggestion. We assume it might be an error from the word processor.

Point 3: References [49], [50] and [51] from lines 235-237 should not be exponential.

Response 3: Thanks for this correction, we have modified the manuscript from line 281 to 285.

Point 4:The conclusion section could be expanded more.

Response 4: In relation to this suggestion, we have expanded the text on lines 351 to 353 in the Conclusions section.

Round 2

Reviewer 1 Report

The authors have adequately addressed the issues I raised!